# Acquired Vitamin B12 Deficiency in Newborns: Positive Impact on Newborn Health through Early Detection

**DOI:** 10.3390/nu14204397

**Published:** 2022-10-20

**Authors:** Patrícia Lipari Pinto, Cristina Florindo, Patrícia Janeiro, Rita Loureiro Santos, Sandra Mexia, Hugo Rocha, Isabel Tavares de Almeida, Laura Vilarinho, Ana Gaspar

**Affiliations:** 1Hereditary Metabolic Disease Reference Center, Metabolic Unit, Pediatric Department, Santa Maria’s Hospital-Lisbon North University Hospital Center, EPE, Pediatric University Clinic, Faculty of Medicine, University of Lisbon, 1600-190 Lisbon, Portugal; 2Laboratory of Metabolism and Genetics, Department of Pharmaceutical Sciences and Medicines, Faculty of Pharmacy, University of Lisbon, 1649-019 Lisbon, Portugal; 3Hereditary Metabolic Diseases Reference Center, Dietetic and Nutrition Department, Santa Maria’s Hospital-Lisbon North University Hospital Center, EPE, Pediatric University Clinic, Faculty of Medicine, University of Lisbon, 1600-190 Lisbon, Portugal; 4Newborn Screening, Metabolism and Genetics Unit, Human Genetics Department, National Institute of Health Dr Ricardo Jorge, 4000-404 Porto, Portugal

**Keywords:** vitamin B12 deficiency, newborn screening, pregnancy, total homocysteine, methylmalonic acid

## Abstract

The early diagnosis of and intervention in vitamin B12 deficiency in exclusively breastfed infants by mothers with low vitamin B12 is crucial in preventing possible irreversible neurologic damage, megaloblastic anemia, and failure to thrive. We assess the usefulness of the early detection of asymptomatic B12 deficiency related to acquired conditions and highlight the importance of monitoring serum vitamin B12 levels during pregnancy. We describe demographic, clinical, dietary, and biochemical data, including the evolution of a vitamin B12 deficiency’s functional biomarkers. We enrolled 12 newborns (5 males) with an age range of 1–2 months old that were exclusively breastfed and asymptomatic. These cases were referred to our metabolic unit due to alterations in expanded newborn screening: high levels of methylmalonic acid and/or total homocysteine (tHcy). All mothers were under a vegetarian diet except three who had abnormal B12 absorption, and all presented low or borderline serum B12 level and high plasma levels of tHcy. Supplementation with oral vitB12 re-established the metabolic homeostasis of the mothers. In infants, therapy with an intramuscular injection of 1.0 mg hydroxocobalamin led to the rapid normalization of the metabolic pattern, and a healthy outcome was observed. Acquired B12 deficiency should be ruled out before proceeding in a differential diagnosis of cobalamin metabolism deficits, methylmalonic acidemia, and homocystinuria.

## 1. Introduction

Vitamin B12 (cobalamin, Cbl), as a regulator of fetal growth, plays an important role in cellular metabolism, affecting cell growth and differentiation by influencing DNA synthesis and epigenetic regulation [1,2]. 

Vitamin B12 deficiency is globally an important public health problem, albeit with still limited population data [3,4]. Some stages of life present a higher risk of deficiency, such as pregnancy and infancy [3,5,6]. Vitamin B12 deficiency has a high global incidence in pregnant women, ranging between 10% and 50% for different populations and ethnicities [7]. In a normal pregnancy, vitamin B12 levels fall by around 30% by the third trimester [5]. Newborn cobalamin levels at birth depend on maternal cobalamin stored during pregnancy, gestational age, and birth weight [6]. Maternal cobalamin deficiency, prematurity, and low birth weight are all associated with lower fetal cobalamin reserves [6]. The risk is higher in exclusively breastfed infants due to low cobalamin intake through the breast milk of mothers with vitamin B12 deficiency [6]. 

Vitamin B12 insufficiency is mainly associated with a high risk of failure to thrive, hematological problems, and short- and long-term effects on neurological and cognitive functions. The extent and degree of disability are assigned to the severity and duration of deficiency status. Moreover, sustained vitamin B12 deficiency may lead to irreversible neurological damage [1,2]. Early diagnosis and intervention in exclusively breastfed infants of mothers with a deficiency in or lack of vitamin B12 supplementation are crucial in preventing possible irreversible neurologic disorders [2].

Vitamin B12 is converted in the mitochondria into adenosylcobalamin (AdoCbl) and in the cytosol into methylcobalamin (MeCbl), which act as cofactors for methylmalonyl-CoA mutase (MCM) and for methionine synthase (MS), respectively. MCM metabolizes the conversion of methylmalonic acid (MMA) into succinyl-CoA, which plays a crucial role in energy metabolism by replenishing the mitochondrial succinyl-CoA pool for the TCA cycle. MS intervenes in the biosynthesis of methionine (Met) through the remethylation of homocysteine (Hcy) using the methyl group donated by the N5-methyl-tetrahydrofolate (Figure 1). Therefore, a decrease in circulating vitamin B12 leads to the disruption of the aforementioned catabolic pathways, resulting in the accumulation of MMA and Hcy. MMA and total homocysteine (tHcy) are used as second-tier test biomarkers in expanded newborn screening (NBS) for the identification of methylmalonic acidurias, defects of intracellular cobalamin metabolism, or elevated Hcy-related disorders. Thus, the alteration in these markers can also be an indication of vitamin B12 deficiency in the neonatal period. The further evaluation of vitamin B12 status via the measurement of serum cobalamin or in some cases holo-transcobalmin, the active form of cobalamin, allows for differentiating the genetic causes of intracellular cobalamin metabolism defects from acquired ones.

We conducted a retrospective study with the aim to assess the usefulness of early therapeutic intervention in asymptomatic infants detected by expanded NBS whose diagnosis of acquired vitamin B12 deficiency was caused by maternal vitamin B12 deficiency. Although our cohort of patients included only 12 cases, it represents a rate of vitamin B12 deficiency of 1 in 29.609 NB, which is like the one described by others [8,9]. We present here the evolution of functional biomarkers from NBS to the first hospital visit and the mothers’ pregnancy outcomes, emphasizing the importance of early diagnosis to prevent the damage of maternal vitamin B12 deficiency.

## 2. Materials and Methods

This study was based on a retrospective observational cohort of newborns referred to the metabolic unit at Hereditary Metabolic Disease Reference Center, Lisbon North University Hospital Center, Lisbon due to alterations in NBS, covering the period from 2017 to 2021. All procedures followed were in accordance with the ethical standards of the responsible committee on human experimentation (institutional and national) and with the Helsinki Declaration of 1975, as revised in 2008.

Demographic data, clinical findings, diet quality and style, and pregnancy follow-up data were collected through a comprehensive review of the individual files. These data were complemented by information obtained during hospital visits. The following biochemical data were compiled: hematological indices, serum vitamin B12, plasma total homocysteine (tHcy), urine methylmalonic acid (MMA), and dried blood spot (DBS) methionine (Met) and propionylcarnitine (C3) levels from first-tier NBS, and tHcy and MMA from second-tier NBS. Functional biomarkers of vitamin B12 deficiency were reassessed in the infants at the first hospital visit with a mean age of 39.2 days (range: 13–66). The biochemical assessment of mothers was also performed at the same visit. The monitoring of functional markers and vitamin B12 serum levels after two weeks of supplementation was subsequently pursued. 

The screening procedure was based on an algorithm that starts with the evaluation of first-tier tests: detection of C3 above the cutoff value (>5.25 µM) or a C3/C2 (acetylcarnitine) > 0.20. Subsequently, in second-tier tests performed on the first DBS (collected between 3rd and 6th days of life), we evaluated MMA, 3-hydroxypropanoic acid, and propionyl glycine. An MMA level above the cutoff value leads to the evaluation of tHcy. 

On the basis of cutoff values usually described in the literature [6], vitamin B12 maternal status was defined as deficient and insufficient for values of vitamin B12 < 200 pg/mL, and between 200 and 300 pg/mL, respectively. Concerning the biomarkers’ normal cutoff values, the ones established by the participant laboratories (personal information) were adopted: dried blood spot (DBS), 99.5% and 1% percentile for high and low values, respectively, C3 (propionylcarnitine) < 5.25 µM; C3/C2 < 0.20; C3/Met < 0.30; MMA < 4.0; tHcy < 4.40 µM and urine MMA < 13.0 µmol/mmol creatinine in infants; plasma tHcy 4.0–6.0 µM, in infants, and ≤14.0 µM for adults [10]. Moreover, megaloblastic anemia was defined for the newborn (NB) when mean corpuscular blood volume (MCV) was >115 fL and hemoglobin < 11.0 g/dL; for adults when MCV was >98 fL and hemoglobin < 11.9 g/dL. Due to the interplay of folates in vitamin B12 deficiency, plasma folate level was evaluated in all mothers and NBs at first hospital visit.

The diagnosis of maternal vitamin B12 deficiency was established when a mother’s vitamin B12 level was below the defined cutoff value and/or functional biomarkers were elevated. In the cases where maternal vitamin B12 deficiency was not explained by dietary reasons, mothers were referred to internal medicine for further work-up and treatment. All mothers were also referred to a nutritional clinic. 

The diagnosis of acquired vitamin B12 deficiency was established in all NBs on the basis of second-tier NBS test results performed on the first DBS and on the data tracked at the first hospital visit: vitamin B12 serum level below cutoff value (<280 pg/mL) concomitant with altered functional biomarker values, the confirmation of maternal vitamin B12 deficiency, and the normalization of NB values after hydroxocobalamin supplementation. 

## 3. Results

In the period of 2017–2021, 12 NBs with altered NBS with suspicion of acquired vitamin B12 insufficiency were studied, and results are presented in Table 1. C3 values varied from high (*n* = 8) to borderline (*n* = 2) and normal (*n* = 2) levels; for Met values, the majority (*n* = 10) were in the normal range, and only two (*n* = 2) displayed Met values slightly below the low range. However, the ratios C3/Met (mean = 0.45; range: 0.23–1.76) and C3/C2 (mean = 0.22; range: 0.13–0.32) for both or one of the two were high. These results led to a second-tier test in the first DBS, and in all cases, MMA (mean = 16.6 µM; range: 6.0–67.8) was above the cutoff level, but for tHcy (mean = 6.9 µM; range: 1.4–16.5), only 7 of the 12 studied cases had a tHcy level above the cutoff value.

The first visit to the hospital occurred at the mean age of 39 days of life (range: 13–66), and the clinical outcome, hematological indices, and functional biomarkers were evaluated (Table 2). All infants were exclusively breastfed and asymptomatic with normal psychomotor development and good weight evolution, except two of them who did not have good weight gain. Four patients included in this study presented transitory anemia with slightly low hemoglobin (9.3 to 10.6 g/dL), and megaloblastic anemia was not detected in any of the infants. 

All NBs revealed very low serum vitamin B12 (<100 pg/mL) and very high levels of tHcy (mean = 71.2 µM; range: 28.0–163.4) (Figure 2), which suggests that the Hcy remethylation pathway under a low vitamin B12 steady state is compromised in a cumulative mode along time. The MMA metabolism seemed to be grossly affected since its urinary excretion was very high on first hospital visit. Met levels were not as informative as could be expected, but the diet also contributes to the cellular Met pool. Plasma Met levels at the first hospital visit were within the respective reference values in all cases (Figure 2).

Therapy with intramuscular vitamin B12 led to the rapid normalization of metabolic abnormalities. At the last visit, all patients with a mean age of one year were asymptomatic with normal levels of the functional biomarkers.

Demographic data, dietary regimen, and biochemical data from the NBs’ mothers are summarized in Table 3. Maternal demographic characteristics show that the affected mothers in our study were from a variety of ethnic backgrounds: Portugal (*n* = 4), India (*n* = 4), Brazil (*n* = 3), and Angola (*n* = 1). Their dietary history and nonvitamin supplementation during pregnancy were potential causes of vitamin B12 deficiency, which was later validated with confirmatory tests. Vitamin B12 deficiency was due to different conditions: decreased intake—nine women adhered to a strict vegetarian diet before and/or during pregnancy; excessive consumption—one was a multipara woman (6th pregnancy) who had multiple vitamin deficiencies; decreased absorption—one was identified with malabsorption syndrome after bariatric surgery; and autoimmune condition—one had a pernicious anemia that had not been previously identified. All mothers showed low levels of vitamin B12 (mean = 137.5 pg/mL; range: <100–267), matching the classification of vitamin B12 deficiency or insufficiency associated with high levels of homocysteine (mean = 18.3 µM; range: 13.4–77.6), and only two displayed slightly high MCV indices, but without anemia (Table 3). The coexistence of folate deficiency was detected only in one case.

Maternal vitamin B12 concentration status was not associated with any perinatal outcomes (birthweight and gestational age at birth, Table 2). Mothers were also supplemented with oral vitamin B12 and folic acid if necessary.

## 4. Discussion

A correctly planned vegetarian diet is nutritionally adequate and healthy [11]. However, nonsupplemented populations who circumvent food of animal origin are at high risk of acquiring a vitamin B12 deficiency or insufficiency status, which relies exclusively on intake from dietary sources [6,10,11,12,13]. Women of childbearing age and pregnant women globally have a high incidence of vitamin B12 deficiency ranging between 10% and 50% for different populations and ethnicities [11,12,13,14].

Vitamin B12 deficiency becomes a relevant health issue specially during pregnancy since vitamin B12 is a relevant regulator of placentation and fetal growth. Vitamin B12 deficiency is frequently underdiagnosed in pregnant women, not only due to a restricted diet, but also due to other causes, as our study shows (excessive consumption, malabsorption, and autoimmune disorders). The most common cause of vitamin B12 deficiency in adults is pernicious anemia [13,15]. However, in our study, only one mother was diagnosed with this condition, showing that a vegetarian diet is likely becoming more common. Several studies have shown a high adherence to vegetarian diets. The global number of vegetarians can represent up to 10% of the total population [11].

Women are frequently clinically asymptomatic in the presence of atrophic gastritis [13]. Pilot project Newborn Screening 2020 (NBS 2020) carried out in Heidelberg, Germany [13,16] performed a diagnostic work-up for the mothers of newborns with vitamin B12 deficiency that led to the diagnosis of previously unrecognized gastrointestinal malabsorption due to autoimmune gastritis, ulcerative colitis, gastric bypass, hemolysis, elevated liver enzymes, and low platelets (HELLP) syndrome, severe pancytopenia, and carbamazepine treatment. However, no gastrointestinal cause for vitamin B12 deficiency was established in most cases and 89% of mothers reported a balanced diet. A vegetarian diet was reported only in a few cases, but some of them had hyperemesis or aversion to meat during pregnancy, which may explain some cases of vitamin deficiency [12,13,16]. Two of the mothers included in our study also developed an aversion to meat or dairy products during pregnancy. It is also important to analyze the fact that 8 of the 12 mothers included in our study were immigrants, showing the possible effect, referred by Reischl-Hajiabadi A.T. et al., of a lack of information, language barriers, or greater hesitancy to attend preventive prenatal care in this group [16].

Vitamin B12 deficiency in newborns is mainly maternal in origin [13,16,17,18]. The best means of preventing neonatal deficiency is to ensure that the mother is vitamin B12-replete during pregnancy and breastfeeding. Women following vegetarian diets should have nutritional counseling, preconception, and monitoring throughout the pregnancy, and start early micronutrient supplementation, including vitamin B12 [1,2,5,11,19]. However, maternity guidelines usually do not include the routine evaluation of vitamin B12 status and vitamin B12 supplementation, so none of the affected mothers in our study had taken vitamin supplementation before or throughout pregnancy. Caregivers of pregnant women should increase awareness of vitamin B12 deficiency. The well-established recommendation and accepted global guidance to start folic acid supplementation preconceptionally may explain the fact that only one of the mothers included in our study showed folate deficiency. 

An additional benefit for NBS may be the detection of undiagnosed vitamin B12-de-ficient NBs from vitamin B12-deficient mothers. The prevalence of vitamin B12 deficiency detected by NBS was described in several studies in the last few years [2,15,16,19,20,21]. The percentage of detected cases varies between countries depending on the strategies and cutoffs applied in each NBS [16].

The MMA was elevated at the time of NBS in all newborns referred to our center, but the same was not observed for tHcy. This observation indicates that MMA is more sensitive than tHcy once the enzymatic reaction involved in MMA catabolism is entirely dependent on vitamin B12 availability. tHcy, as a biomarker of vitamin B12 deficiency, may be ambiguous since its metabolism is interconnected with folate. However, after a short time (1 or 2 months), plasma tHcy was significantly high. The rapid rise in biomarkers with cellular adverse effects emphasizes the impact of an early therapeutic intervention towards a healthy outcome. This point is controversial; in some studies [12], the most sensitive marker for vitamin B12 deficiency was tHcy, but it is the consensus that the combination of MMA and tHcy is needed in all cases. 

There is no agreement regarding treatment in cases of vitamin B12 deficiency [13]. To all NBs included in this study, a one-time injection of 1.0 mg of hydroxocobalamin improved the metabolic status. Pilot project NBS 2020 [12,16] compared the effectiveness of oral-only vitamin B12 supplementation versus supplementation and parental therapy (including intramuscular and intravenous), showing that vitamin B12 levels and functional biomarkers were normalized with both treatments. The group treated with parenteral supplementation showed a higher response in terms of vitamin B12 levels, which may indicate excessive treatment due to the achieved supranormal vitamin B12 levels [16]. In another study, 47 children with vitamin B12 deficiency were treated with oral vitamin B12, showing its effectiveness [22]. Oral supplementation seems to be a good treatment option to avoid invasive and painful treatments in families with compliance to the treatment. Vtamin B12 deficiency in the child or mother should not be a reason to avoid breastfeeding rather than adequately monitoring and supplementing vitamin B12 [13]. 

Several studies [5,19,23,24,25,26] demonstrated that cobalamin deficiency in exclusively breastfed infants is essentially manifested between 3 and 6 months of age, and in older ages in strict association with the decline in a breast-milk cobalamin source. Symptoms usually start between 4 and 6 months of age if vitamin B12 levels are not corrected [13], and differ in severity, including physical, hematological, and neurological signs. All patients of our study remained without clinical symptoms at a one-year follow-up, which was expected, since early therapeutic intervention may prevent neurological and severe hematological findings. 

This supports the postulated clinical benefit achieved by NBS and the consequent early treatment of still-asymptomatic children due to maternal vitamin B12 deficiency under exclusive breast-milk feeding. The percentage of cobalamin in breast milk is a major determinant of cobalamin status in exclusively breastfed infants and is strongly correlated with maternal blood cobalamin [19]. Maternal cobalamin intake is the main determinant of a breast-milk cobalamin source [19]. Therefore, most commercially prepared infant formulas are enriched with cobalamin to overcome the problem. Moreover, maternal cobalamin deficiency was related to an increased risk of preterm birth and low birth weight [6,7,12,13,23,25]; however, in our study and pilot project NBS 2020 [16], this correlation was not observed. 

## 5. Conclusions

In our study, early detection through NBS may have prevented hematological abnormalities and irreversible neurological damage in infants with acquired vitamin B12 deficiency due to maternal vitamin B12 deficiency. Cobalamin deficiency after birth is a global public health problem, and a serious one in countries with endemic deficiency and prevalent or prolonged breastfeeding practices. Future improvements in the feedback from metabolic centers that are responsible for the ultimate diagnosis of these conditions to NBS follow-up programs result in better knowledge of the incidence of this preventable nutritional condition [2,27].

The most effective prevention strategy for both mother and child includes screening for maternal vitamin B12 deficiency throughout the pregnancy on follow-up appointments [12,16]. Healthcare providers should ask pregnant and lactating women about their diet and medical history to identify those who are at risk of an inadequate intake or malabsorption of vitamin B12. Providers should not rely solely on the measurement of serum vitamin B12 level but should cross-reference these data with vitamin B12 functional marker measurements, plasma MMA, and tHcy to confirm the status of vitamin B12 deficiency in at-risk women [2]. Caregivers of pregnant women should be aware of vitamin B12 deficiency and prevent it during pregnancy. 

## Figures and Tables

**Figure 1 nutrients-14-04397-f001:**
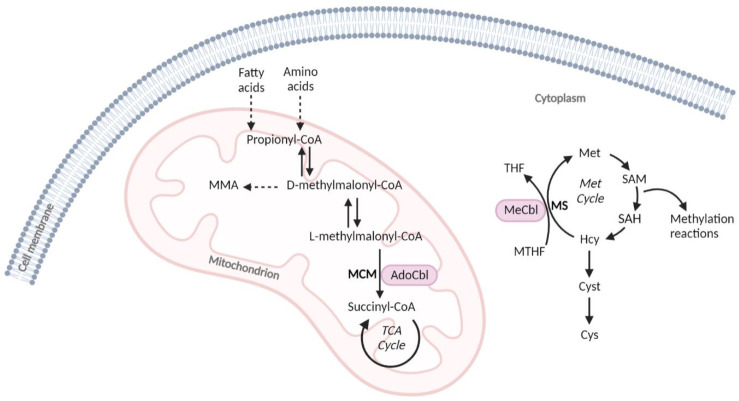
Simplified scheme of the two metabolic pathways that use cobalamin cofactors. AdoCbl, adenosylcobalamin; Cys, cystine; Cyst, cystathionine; Hcy, homocysteine; MeCbl, methylcobalamin; Met cycle, methionine cycle; MCM, methylmalonyl-CoA mutase; MMA, methylmalonic acid; MS, methionine synthase; MTHF, N5-methyltetrahydrofolate; SAH, S-adenosylhomocysteine; SAM, S-adenosylmethionine; TCA cycle, tricarboxylic acid cycle; THF, tetrahydrofolate.

**Figure 2 nutrients-14-04397-f002:**
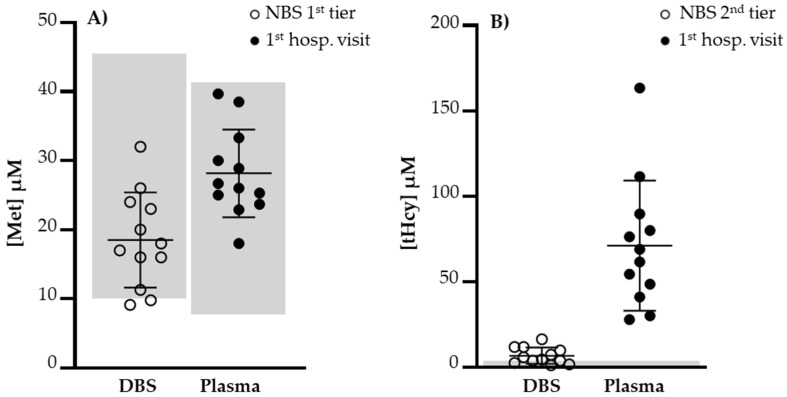
(**A**) Levels of DBS Met at first NBS tier and plasma Met at first hospital visit. Met values did not reflect a compromise of remethylation reaction due to vitamin B12 deficiency. (**B**) Levels of DBS tHcy at second NBS tier and plasma tHcy at second hospital visit. A significant increase in tHcy was observed at first hospital visit, 32.9 days of age (average, range 13–65). Met, methionine; NBS, newborn screening; tHcy, total homocysteine. Normal value ranges limited by gray areas.

**Table 1 nutrients-14-04397-t001:** Infant birth outcomes and DBS biomarkers at first and second NBS tiers.

		Birth Outcome		DBS Biomarkers First NBS Tier				DBS Biomarkers Second NBS Tier	
Index Case	Sex	GAB (Weeks)	BW (g)	C3 (µM)	C3/C2	Met (µM)	C3/Met	MMA (µM)	tHcy (µM)
GJ	M	41	3220	5.03	0.25	16.0	0.31	7.3	2.8
PD	M	38	3125	5.07	0.17	9.8	0.52	10.4	10.0
DG	M	39	3130	16.05	0.32	9.1	1.76	67.8	12.1
AS	M	40	2580	3.67	0.21	16.0	0.23	10.9	16.5
WC	M	41	3705	9.41	0.22	17.0	0.55	14.0	6.0
ACS	F	40	2885	5.73	0.24	18.0	0.32	20.0	7.5
KS	F	40	3180	4.61	0.22	20.0	0.23	16.4	4.8
SP	F	37	3005	5.33	0.18	32.0	0.17	6.0	1.9
IQ	F	41	3195	6.03	0.25	11.3	0.53	16.5	12.0
KB	F	39	3035	5.52	0.23	24.0	0.23	11.6	4.1
CS	F	40	3600	5.43	0.18	26.0	0.21	10.7	4.1
SK	F	39	3105	7.67	0.13	23.0	0.33	7.0	1.4
		Cutoff values:		<5.25	<0.22	10.0–45.0	<0.30	<4.0	<4.4

GAB, gestational age at birth; BW, birthweight; DBS, dried blood spot; NB, newborn; NBS, newborn screening. C2, acetylcarnitine; C3, propionylcarnitine; Met, methionine; MMA, methylmalonic acid; tHcy, total homocysteine. Cutoff values: 99.5% and 1% percentile for high and low values, respectively.

**Table 2 nutrients-14-04397-t002:** Infant outcomes and evaluated biomarkers at first hospital visit.

	Infant Outcome		Hematological Indices			Biomarkers				
						Serum		Plasma		Urine
Index Case	First Hospital Visit (Age in Days)	Weight (g)	Hb (g/dL)	MCV (fL)	Hct (%)	Folate (nM)	Vit B12 (pg/mL)	Met (µM)	tHcy (µM)	MMA (µmol/mmolCrn)
GJ	51	3900	12.2	92.7	34.4	15.4	<100	26.0	54.5	183.0
PD	26	4060	10	103	28.0	>20	<100	22.9	111.5	605.0
DG	13	2858	13.4	94.9	58.9	>20	<100	39.7	163.4	NA
AS	51	3864	10.6	90.8	30.5	>20	<100	33.3	89.8	NA
WC	30	4910	11.4	89.2	32.9	>20	128	25.0	41.3	281.0
ACS	20	3116	13.5	95.8	39.3	>20	<100	26.7	76.4	510.8
KS	24	NA	17.9	101.6	54.0	>20	<100	28.9	61.7	NA
SP	66	4468	9.8	NA	33.7	NA	202	25.3	28.0	NA
IQ	26	3968	12.6	92.3	36.5	>20	<100	18.0	80.0	26.2
KB	65	4835	11.1	88.9	31.0	12.2	<100	30.0	69.0	NA
CS	40	4900	9.3	95.7	26.1	>20	<100	23.7	48.8	825.8
SK	58	3900	11.2	80.4	32.7	>20	<100	38.5	30.3	261.4
	Reference values:		>11.0	<115	32–42	>6.1	>285	8.7–40.9	4.0–6.0	<13.0

Hb, hemoglobin; MCV, mean corpuscular blood volume; Hct, hematocrit; Met, methionine; MMA, methylmalonic acid; tHcy, total homocysteine; NA, not available.

**Table 3 nutrients-14-04397-t003:** Maternal demographic and clinical findings, dietary regimen, and biochemical data.

	Maternal Data				Hematological Indices			Plasma Biomarkers		
Index Case	Age (Years)	Origin	Clinical Status	Dietary Regimen	Hb (g/dL)	MCV (%)	Hct (%)	Folate (nM)	Vit B12 (pg/mL)	tHcy (µM)
GJ	29	PT	-	Vegetarian	13.2	96.0	39.0	9.8	168	24.0
PD	35	PT	Anemia *	-	13.2	103.4	37.7	12.6	<100	32.7
DG	38	BR	-	Vegetarian	13.4	94.9	39.5	>20	<100	NA
AS	32	PT	-	Restriction of meat and milk during pregnancy	10.4	74.9	31.2	12.3	<100	NA
WC	36	AO	Malabsorption syndrome	-	12.9	89.4	37.3	>20	108	77.6
ACS	34	PT	-	Restriction of dairy products during pregnancy	14.3	100.2	42.7	>20	<100	49.3
KS	25	IN		Vegetarian	12.7	96.6	39.8	6.4	<100	19.4
SP	30	IN	-	Restriction of dairy products and meat	12.3	76.7	38.6	7.6	267	13.4
IQ	36	BR	Pernicious anemia	-	12.1	91.2	39.2	>20	<100	50.3
KB	32	IN	-	Vegetarian	11.4	79.5	35.5	3.25	<100	NA
CS	40	BR	-	Restriction of meat and eggs	12.7	84.5	38.2	>20	146	22.9
SK	28	IN	-	Vegetarian	13.1	78.8	38.8	6.2	129	13.7
				Reference values:	>11.9	<98	35.4–44.4	>5.4	>200	<14.0

Hb, hemoglobin; MCV, mean corpuscular blood volume; Hct, hematocrit; tHcy, total homocysteine; NA, not available; AO, Angola; BR, Brazil; IN, India; PT, Portugal. * Grand multipara, 6th pregnancy.

## Data Availability

The data presented in this study are available in [Table 1, Table 2 and Table 3 of this article]. More data is available on request from the corresponding author.

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
