# Peer review of "Acquired Vitamin B12 Deficiency in Newborns: Positive Impact on Newborn Health through Early Detection"

_nutrients, 2022, doi:10.3390/nu14204397_

Round 1
Reviewer 1 Report
This manuscript describes newborn screening and confirmatory diagnoses of vitamin B12 deficiency. The condition potentially causes severe neurological damages that can be prevented by early detection and supplementaion of vitamin B12. However, C3-acylcarnitine appears to be less sensitive for vitamin B12 deficiency and disorders of cobalamin metabolism than for methylmalonyl-CoA mutase deficiency. Therefore, the data presented in this article will be very useful for those who are interested in newborn screening for these disorders. My questions and comments are as follows.
1)
How many newborns were screened within the study period, and how many of them were positive for C3? Is it possible to show the numbers (or ratios) of methylmalonyl-CoA mutase deficiency, disorders of cobalamin metabolism, propionic acidemia, and false-positive cases as well as vitamin B12 deficiency?
2)
Explain how the cutoff values for the following indices in DBS were set: C3, C3/C2, Met, C3/Met, MMA, tHcy. (e.g. Cutoff for C3 as 5.25 corresponds to mean + #sd (or to ## percentile, etc.).
3)
There are the same reference values for Met and tHcy shown in Table 1 (in DBS for screening) and Table 2 (at the first visit). Were the data in Table 2 also values in DBS? Please clarify the type of samples used.
Author Response
Please see the attachment.
We really hope to have answered to all pertinent raised questions by the reviewers and that now the manuscript has reached the intended standard for publication.
With our kindest regards,
Patrícia Lipari Pinto

Reviewer 2 Report
The manuscript entitled "Acquired vitamin B12 deficiency in newborns: positive impact on newborn health through early detection" discuss the importance of vitamin B12 in both newborns and mother.
The manuscript is well written but the author should consider the following challenges
1. The data pool is very small and concluding from these data is not enough.
2. There are several reports on a similar topic for example
Current Medical Science 40, 801–809 (2020)
3. Authors need to include more references and extensively discuss the reported data and present data to have a better conclusion.
4. At this point the discussion section is underdeveloped.
Author Response

(The authors gave the same response as above.)

Round 2
Reviewer 2 Report
The revised manuscript is now suitable for publication.
Author Response
Dear Sir,
We acknowledge the comments as well as the opportunity to submit a reviewed version of the manuscript.
The comments were pertinent, and they were useful to improve the manuscript content.
Thank you for your approval.
Best regards,
Patrícia Lipari Pinto